# Single cell mapping identifies a distinct platelet-phenotype in psoriatic type III inflammation

Katharina S. Kommoss [1,2,14], Sinduya Krishnarajah [3,12,14], Tabea Bieler [1,4], Ekaterina Friebel [3,13], Lukas Rindlisbacher[3], Pierluigi Ramadori [1], Philipp Häne [3], Shing Kam[1], Florian Müller[1], Sandra Prokosch[1], Ulrike Rothermel[1], Yunchen Wu[5], Francesca Barletta [6,7], Stefan Czemmel [6,7], Sven Nahnsen [6,8,9], Dominic Helm [10], Martin Schneider[10], Knut Schäkel [2], Alexander Enk [2], Burkhard Becher [3,15] ✉ & Mathias Heikenwälder [1,9,11,15] ✉

The understanding of platelet biology is expanding beyond their well-established role in thrombosis and hemostasis to encompass their functions in inflammation. To gain insights into the phenotype and possible platelet-subgroups in the context of type III inflammation, high-parametric single cell spectral flow cytometry was applied to a prospectively followed cohort of psoriasis patients undergoing systemic therapy. By focusing on the activation profile of platelets and their interaction with immune cells, we identify cell-surface proteins CD32+CD154+ and TLR2+TLR4+, distinguishing unique platelet subsets in psoriasis patients. These subsets demonstrate regression over the course of systemic therapy. Notably, these platelet phenotypes and frequencies differ from those of healthy controls, and patients with atopic dermatitis (AD), a type II inflammation-driven disease. Our data highlight a specific platelet phenotype responding acutely to the respective inflammatory context. This finding warrants further investigation into platelets as a diagnostic tool in inflammatory conditions, necessitating future insight into the function of these subsets across disease etiologies.

Platelets are small, anucleate cells derived from megakaryocytes[1]. Their relatively short life span of 5–12 days reflects the acute nature of their physiological functions. Constantly patrolling the circulatory system, platelets are highly responsive to vascular injury and external stimuli, rapidly altering their surface receptors upon activation. This activation is further enhanced through autocrine and paracrine signaling, leading to the recruitment of additional immune cells to sites of injury and inflammation. Whilst platelets are traditionally recognized for their well-established role in thrombosis and hemostasis, there is growing evidence that platelets play a central role in systemic inflammatory processes[2]. Platelets in principle have the capacity to orchestrate inflammation either through cell contact, or via secretion of cytokines and chemokines[3–5]. As such however, platelet signatures in disease are an emerging area of research with direct implications in clinical translation for diagnostics and therapeutic target identification. Specifically, phenotypic diversity of platelets in inflammatory diseases remains underexplored. Recent advances in single-cell technologies (i.e. high dimensional cytometry by time of flight[6], single-cell RNA sequencing[7], single-cell proteomics[8]), offer powerful tools to uncover distinct platelet signatures, opening new avenues for diagnostic and translational research.

Inflammation can broadly be classified into three overarching types based on leukocyte subtypes (e.g., T-helper 1 (Th1)/Th2/Th17 cells) and their associated cytokines (e.g., IFN-gamma/IL-4/IL-17)[9,10]. In

---

this study, we focus on type III inflammation by investigating platelet signatures in psoriasis, a prototypical type III-inflammatory condition.

In this work, we investigate platelet surface markers in psoriasis, a classical type III-inflammatory disease, to identify the signatures of platelets in the context of inflammation with single-cell resolution[9,11]. Within a prospective, single-center study (Fig. 1a), platelets derived from psoriasis patients are followed from maximum dermal inflammation (baseline) to subsequent reduction after 1 and 4 months of systemic therapy. This enables dynamic examination of platelet-phenotype in the context of inflammation on an intra-individual level. Platelet-phenotype is compared to healthy controls, as well as a comparative cohort of patients with a classical type II inflammatory disorder, atopic dermatitis (AD)[11].

## Results

Volunteers for blood sampling were recruited in three cohorts from a single center in Heidelberg (Germany). In addition to a healthy control group, patients with psoriasis or AD were followed prospectively for up to 4 months of systemic dermatological therapy. Patients donated blood samples at three distinct timepoints—baseline; 1 month and 4 months post- systemic therapy. This enabled continuous tracking of platelet phenotypes and their inflammatory state from peak disease severity through initial treatment response and subsequent resolution of skin inflammation.

The extent and severity of the disease-specific skin inflammation was measured using clinical scores (psoriasis area and severity index (PASI)/eczema area and severity index(EASI)), both ranging from 0 to 72 points. These values were similar at baseline for psoriasis and AD patients (PASI 17.19 ($\pm$9.53); EASI 19.93 ($\pm$6.30)), although these scores assess distinct features of the different diseases (erythema, thickness, desquamation for PASI vs. erythema, thickness, excoriation and lichenification for EASI) and therefore a direct comparison is not entirely precise. However, both groups showed a similar response to therapy ($\Delta$PASI of 15.98 ($\pm$9.30); and $\Delta$EASI of 13.67 ($\pm$6.22)) after 4 months of treatment (Supplementary Table 1). To systematically examine the nature of platelets present in the blood of these patients, we applied high-dimensional flow cytometry using a panel designed to identify surface and intracellular proteins associated with pan-platelet characterization, activation, trafficking, and platelet-leukocyte interactions at single-platelet resolution (Fig. 1a, Supplementary Table 2).

A conventional, biased data analysis approach limited to current knowledge in the literature may not fully capture disease-related phenotypic changes. However, data-driven approaches offer new possibilities to identify intrinsic clinical trends, although their translational value can be limited by their complexity. To address this, we combined both hypothesis-driven and data-driven approaches to ensure relevant platelet subsets were identified, and these findings were validated in two additional independent cohorts.

Initial principal component (PC) analysis projected all analyzed platelets from the entire cohort, confirming distinct platelet phenotypes separating psoriasis from AD and healthy controls at baseline (Fig. 1b; Supplementary Table 3). A scree plot defined the contribution of individual PCs in data skewing, identifying the first two components as the main drivers of data separation (Fig. 1c). Key markers in these two components included CD32, CD40-ligand (CD154), and toll-like receptors (TLR) 2 and 4. Interestingly, these trends were most prominent at baseline, diminishing with advancing treatment (Fig. 1d), suggesting that successful dermatological therapy leads to a resolution of platelet activation and inflammation. Taken together, the initial analysis of data trends indicates the presence of disease- and disease severity- related platelet phenotypes.

To gather an overview of the platelet subpopulations present in the systemic immune compartment, flow cytometry data across the three timepoints were visualized using Uniform Manifold Approximation and Projection (UMAP)[12] and subsequently clustered using FlowSOM[13], yielding phenotypically distinct islands of platelets at each timepoint. Manual annotations were assigned to each cluster identified by FlowSOM using the expression of combined markers. Using this approach, we were able to map platelet subsets to serve as the basic substrate for data-driven analysis (Fig. 1e–g).

Specifically, the combined expression of $CD32^+CD154^+$, $TLR2^+TLR4^+$ and $GARP^+P2Y12^+$ generated distinct platelet subpopulations, as shown by the representative mean fluorescence intensities (MFI) of these markers compared to negative control (Fig. 2a). Quantification of the clusters within the three cohorts demonstrated frequencies of $CD32^+CD154^+$ and $TLR2^+TLR4^+$ significantly increased in psoriasis patients compared to healthy controls ($p = 0.048$ and $p = 0.049$, respectively) at baseline, as well as $GARP^+P2Y12^+$ after four weeks of therapy ($p = 0.042$; Fig. 2b). Of note, these clusters were not increased in the AD-cohort. Importantly, no correlation between body mass index (BMI) and frequency of the $CD32^+CD154^+$ cluster was observed in correlation analyses at baseline, and differences remained significant after correction for BMI in a multiple linear regression analysis and matching for BMI < 25 (Supplementary Fig. 1a–c). Manual gating confirmed the data-driven findings, showing a decrease in the $CD32^+CD154^+$ cluster in psoriasis patients over time, approaching control signatures (Fig. 2c, d), corroborating the findings generated by the unbiased approach. Correlation analysis for the individual patients revealed a trend for positive correlation for $CD32^+CD154^+$ frequencies with PASI, reflecting clinical extent and severity of skin involvement (Fig. 2e). However, this was not observed for more generic markers of systemic inflammation (such as white blood count and c-reactive protein; Supplementary Fig. 1d, e).

Concurrent total proteome analysis of bulk platelet samples collected from the investigated patients/donors at the same timepoint did not reveal statistically significant differential expression of the markers discussed above (Supplementary Fig. 2, Supplementary Fig. 3, Supplementary Table 4-5). However, from 2611 identified proteins, other 35 and 4 differentially expressed proteins aside from the investigated surface markers were demonstrated for psoriasis (Supplementary Fig. 2a–c, Supplementary Fig. 3a–c, Suppl. Table 6) and AD (Supplementary Fig. 2d–f, Supplementary Fig. 3d–f, Supplementary Table 7) patient platelets in comparison to healthy controls, respectively. Additional differentially expressed proteins were identified under investigation for therapy timeline (Supplementary Fig. 2g–l, Supplementary Fig. 3g–l), and in comparison of psoriasis and AD platelets at baseline (Supplementary Fig. 2m, Supplementary Fig. 3m). Taken together, the bulk proteomic platelet analysis indicated no trackable platelet-phenotype specific to psoriasis or AD. This highlights the specificity of the combinations found using high-dimensional flow cytometry, as well as the necessity to use single cell technologies to uncover inflammation-type related differences.

To validate the findings, two additional patient cohorts were analyzed using spectral flow cytometry (Supplementary Table 8, 9). The increase of $CD32^+CD154^+$ platelets in psoriasis patients versus healthy controls retained the statistical significance in both cohorts (cohort 2, $p = 0.041$, Fig. 2f; cohort 3, $p = 0.006$, Supplementary Fig. 1g). The validation cohorts again supported the trend in the increase of $TLR2^+TLR4^+$ and $GARP^+P2Y12^+$ platelet frequencies (Supplementary Fig. 1f–g). Importantly, AD patients demonstrated frequencies similar to those of healthy controls in both cohorts, corroborating the findings in cohort 1. The overall findings are schematically summarized in Fig. 2g.

## Discussion

As our understanding of the complex subgroups or types of inflammation grows, there is a rising unmet need to investigate key players in inflammation beyond leukocyte subgroups and cytokines. Specifically, our knowledge of the role of platelets is evolving from their traditional functions in hemostasis and thrombosis, towards a central role in inflammation[2]. Platelet phenotyping has the potential to reveal dynamic changes in potential subsets of platelets which have been

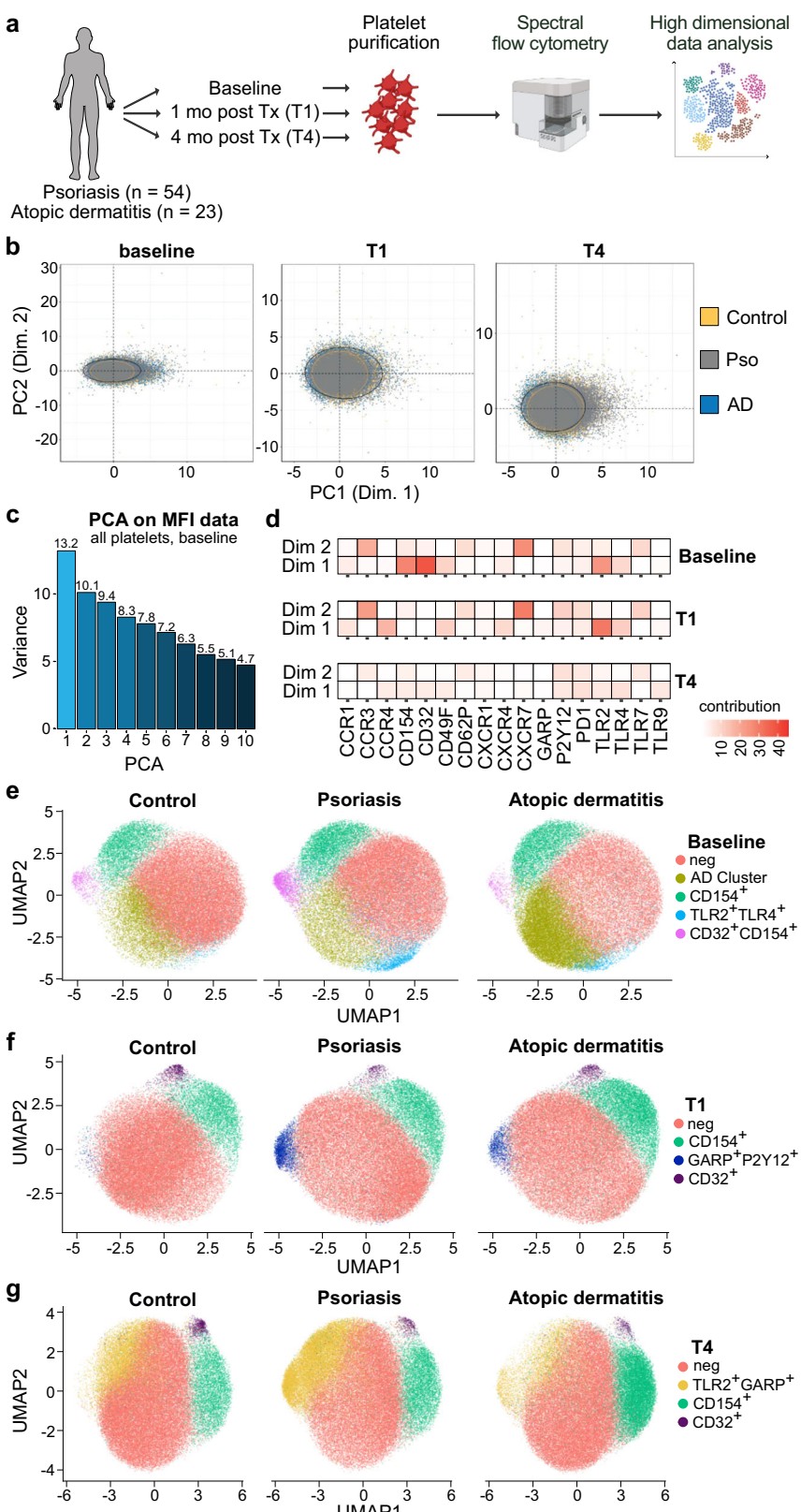

thus far overlooked and consecutively offer diagnostic and therapeutic value in inflammatory diseases. In this study, we focused on platelet surface markers that reflect platelet activation and interaction with both innate and adaptive immune cells.

Studying platelets in inflammatory dermatoses, such as psoriasis (type III inflammation) and atopic dermatitis (AD, type II inflammation), allows for easy clinical assessment of dermal inflammation using well-established indices like the psoriasis area and severity index (PASI) and the eczema area and severity index (EASI). Our longitudinal study over four months enabled the specific investigation of skin inflammation-related changes in platelets, as other possible confounders (i.e. body mass index, arterial hypertension, smoking status etc.) remained predominantly unchanged in individual patients. Therefore, the changes in frequencies observed in this study are likely

**Fig. 1 | Unbiased data analysis reveals platelets display distinct type-III specific phenotype. a** Schematic overview over study design. Psoriasis (pso) and atopic dermatitis (AD) patients were prospectively followed from baseline, to one month post systemic therapy (Tx) initiation (T1), to 16 weeks post systemic therapy initiation (T4). At each timepoint, platelets were purified from whole blood and subjected to spectral flow cytometry analysis. Created in BioRender. Heikenwälder, M. (2025) https://BioRender.com/jvyk1jp. **b** Unbiased principal component (PC) analysis of the total portion of analyzed platelets from the complete cohort 1.

**c** Scree plot of principal component analysis (PCA) on mean fluorescence intensity (MFI) data of all platelets at baseline. **d** Heatmap depicting the contribution of the individual markers of the first two components of the PC Analysis. Visualization of the data, stratified by timepoint into baseline (**e**), T1 (**f**) and T4 (**g**), using Uniform Manifold Approximation and Projection (UMAP), and subsequently clustered using FlowSOM. Data shown represents the analysis of cohort 1 ($n = 10$ controls, $n = 27$ psoriasis patients, $n = 4$ AD patients), which was processed, measured and analyzed in one batch. Source data are provided in a source data file.

due to the rapid and vast changes in skin type III/II-inflammation, reflected by altered PASI/EASI scores. Emerging evidence supports an inflammatory role for platelets in chronic-inflammatory dermatoses, with platelets modulating inflammation in both psoriasis and AD (reviewed by Tamagawa-Mineoka[14]).

The concept of "types" of inflammation with regards to Th-cell subsets[9,10], is based on associated cellular interaction partners and cytokines. Many studies on the inflammatory role of platelets have focused to date on their interactions with specific cellular subtypes such as neutrophils[4,15], dendritic cells,[16] monocytes/macrophages[5,17], and T-cells[18,19]. Additionally, it has been repeatedly reported that platelets display an "activated" phenotype across different types of inflammation, measured through CD62P expression. However, studies on overarching "platelet phenotypes" in the context of specific inflammatory subtypes are lacking.

In our study, a distinct platelet phenotype was observed in psoriatic type III inflammation but not in AD (type II inflammation) (Fig. 2g). The markers defining the observed type III platelet phenotype are connected to type III inflammatory cytokines (i.e. CD154 – CCL2, IL-8[20]; CD32 - interferon gamma[21]; TLR2 – RANTES, sCD40-ligand[22]) and cellular interaction partners (i.e. TLR4 - neutrophils[15]).

CD154, also known as CD40 ligand (CD40L), is a member of the TNF superfamily that triggers inflammatory responses —such as the production of IL-1, IL-6, and TNF—through interaction with CD40 on B cells, monocytes, macrophages, dendritic cells, and endothelial cells[23]. Platelets store preformed CD154, which can be rapidly expressed on their surface upon activation[20]. In contrast to the de novo synthesis of TNF by platelets—which functions in both its transmembrane and soluble forms—CD154 signaling is faster and becomes active only upon engagement of CD40, particularly on monocytes and endothelial cells. This suggests that CD154 may act as a rapid signal amplifier in psoriatic inflammation, primarily driven by type III inflammatory leukocytes.

CD32, which is constitutively expressed on platelets, enhances platelet activation and has been shown to be upregulated by interferon-gamma[21]. This regulation may account for the increased CD32 expression observed in psoriasis patients at baseline compared to healthy controls and individuals with atopic dermatitis.

Furthermore, platelet-expressed TLR4 has been shown to activate neutrophils and promote the formation of neutrophil extracellular traps (NETs) in the context of sepsis[15]. Given that NET formation is a hallmark of psoriasis[24], TLR4 on platelets may contribute to the self-sustaining inflammatory loop characteristic of the disease. In addition, TLR2 activation on platelets has been shown to induce the release of soluble CD62P, RANTES, and sCD40L[22], further fueling a feed-forward cycle of type III inflammation.

Despite reports on platelet involvement with leukocytes and cytokines in type II inflammation (i.e. IL-33 expression and modulation of eosinophils[25], platelet-eosinophil interaction[26]), no distinct type II-phenotype was apparent in the investigated AD cohorts. This supports the notion that platelets have a greater influence on type III than type II inflammation, warranting further investigation into the functional impact of these platelet subsets.

In clinically challenging cases, i.e. erythroderma, current approaches for diagnostic distinction are mainly based on skin biopsies and both time- and cost-consuming (i.e. single-cell RNA sequencing[27], spatial transcriptomics[28], bulk NanoString RNA-sequencing[29]). Here,

investigation of inflammation-specific platelet profiles could provide a less invasive, and more efficient diagnostic alternative.

Further, it was recently shown that COX-1 inhibition with Aspirin suppressed endothelial cell inflammation in psoriasis patients mediated by hyperactive platelets[30]. Targeting platelet-immune cell interactions, anti-CD154 antibodies have been investigated in type I inflammatory diseases, showing promising results in systemic lupus erythematodes[31] and multiple sclerosis[32]. Understanding specific platelet phenotypes in various inflammatory conditions, along with their downstream functional mechanisms, provides a promising foundation for precise therapeutic interventions. This is especially important because all platelet-targeted treatments carry a risk of bleeding complications.

Limitations of the study include the relatively small sample size of platelet donors, which may reduce the statistical power of our findings. Additionally, it is possible that systemic effects, beyond the reduction of PASI/EASI through administered therapies, may have influenced the observed changes. Potential batch effects in the data analysis also cannot be entirely ruled out. The surface marker panel used in this study was assembled based on a review of literature on platelet activation and immune interactions, but it may not comprehensively capture the breadth of platelet phenotypes involved in inflammation. Lastly, while this study focused on a single representative model for type III and type II inflammation, future research across a broader spectrum of disease, including non-dermatological conditions, will be instrumental in corroborating and extending these findings within diverse inflammatory contexts.

Taken together, our findings demonstrate that distinct platelet subtypes exist depending on the context of inflammation, as evidenced by surface marker expression. This opens avenues for future research into the pathophysiological nature of these subsets in driving disease pathology, and their diagnostic and therapeutic potential in inflammatory diseases.

## Methods
### Human samples
Participants were included after written informed consent, and no compensation was provided. Sex and gender was not considered in the study design, and no sex- or gender-based analyses were performed, as they were determined not relevant within this study setting. Sex was reported as self-indicated in clinical reports. Gender identity was not specifically inquired within this study, as no clinical relevance of this was apparent within this study setting. Clinical routine data and blood samples for platelet isolation were collected in the in- or outpatient clinic of the Department of Dermatology in Heidelberg, Germany. Clinically proven psoriasis or AD patients, diagnosed by board-certified Dermatologists, were included. Patients were eligible for systemic anti-inflammatory therapy according to national and European guidelines[33,34]. Eligibility was defined as either being naïve to systemic therapy or having undergone a 12-week washout of prior therapy. Exclusion criteria consisted of infectious/malignant/additional autoimmune diseases, age <18 years, pregnancy as well as use of anti-platelet or anti-coagulant drugs. Clinical information and blood samples were collected at baseline (before therapy), at one and four months post-initiation of systemic therapy. Therapies such as ixekizumab, secukinumab, brodalumab, guselkumab, and others were administered according to drug data sheets and guideline

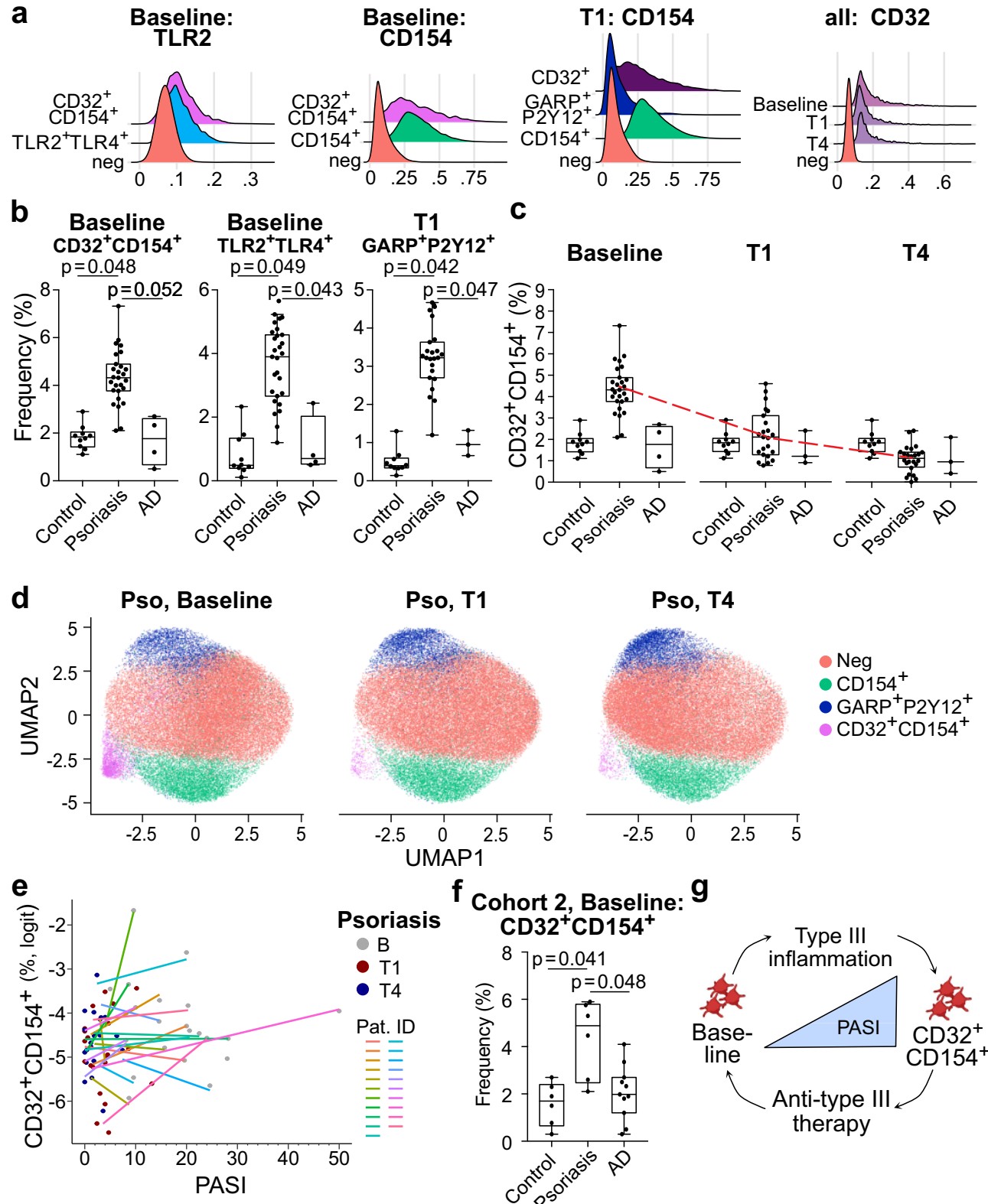

recommendations, considering patient-specific characteristics and decisions made jointly by the patient and physician. Healthy donors were defined as individuals without inflammatory skin diseases and without meeting any of the exclusion criteria.

**Platelet isolation**

Platelets were isolated according to a previously published protocol for platelet RNA sequencing from whole blood collected in EDTA-

coated tubes (cat. no. 02.1066.001, Sarstedt, Nuembrecht, Germany; 3 ml for high dimensional flow cytometry, 9 ml for proteomic analyses), yielding highly purified platelets[35]. In short, within max. 3 h post sample collection, platelet-rich-plasma was collected after a first centrifugation step (120 G, 20 min, room temperature (RT)). For spectral flow cytometry, after a further centrifugation step (120 G, 20 min, RT), the platelet pellet was resuspended in the patient's own plasma containing 5% DMSO (cat. no. D4540-100ml, Merck KGaA, Darmstadt,

**Fig. 2 | Data driven analysis corroborates distinct type-III specific phenotype of platelets. a** Mean fluorescence intensities (MFI) of TLR2, CD154 and CD32 across different timepoints. **b** Quantification of selected clusters at indicated timepoints across healthy controls (control), psoriasis (pso), and atopic dermatitis patients (AD). Statistical analysis was carried out using one-sided unpaired student's *t* tests. Depicted *p* values under 0.05 are considered significant with a Benjamini-Hochberg (BH) false discovery rate (FDR) <5%. **c** Follow-up of CD32⁺CD154⁺ platelet frequencies from baseline to one month (T1) and four months (T4) post systemic therapy initiation. **d** Uniform Manifold Approximation and Projection (UMAP), subsequently clustered using FlowSOM, of psoriasis patient platelets over baseline, T1, and T4. **e** Individual correlation of CD32⁺CD154⁺ platelet frequencies (logit transformed) with psoriasis area and severity index (PASI) in psoriasis patients.

Data shown in (**a**–**e**) represents the analysis of cohort 1 (*n* = 10 controls, *n* = 27 psoriasis patients, *n* = 4 AD patients), which was processed, measured and analyzed in one batch. **f** Frequency of CD32⁺CD154⁺ platelets in the validation cohort 2, which was processed, measured, and analyzed in a separate second batch (*n* = 6 controls, *n* = 6 psoriasis patients, *n* = 10 AD patients). Statistical analysis was carried out using one-sided unpaired student's *t* tests. Depicted *p* values under 0.05 are considered significant. **g** Summarizing overview of findings in this study, Created in BioRender. Heikenwälder, M. (2025) https://BioRender.com/5m1qd4y. Source data are provided in a source data file. For all depicted box plots in this figure, the horizontal line within each box indicates the median, while the lower and upper box edges denote the first and third quartiles, respectively. Whiskers extend to 1.5 times the interquartile range, and dots represent individual data points.

Germany), and transferred to −80 °C in an isopropanol freezing container (Mr. Frosty, cat. no. 5100-0001, Thermo Scientific, Massachusetts, USA). For total platelet proteomics, after a further centrifugation step (120 G, 20 min, RT), the platelet pellet was resuspended in RIPA buffer (cat. no. 9806S, Cell Signaling Technology, Massachusetts, USA) with 1 tablet PHOSSTOP (cat. no. 4906837001 Roche Products, Basel, Switzerland), and cOmplete, Mini, EDTA-free Protease (cat. no. 4693159001, Roche Products, Basel, Switzerland), and directly transferred to −80 °C. Random assessment of platelet purity was performed by microscopy (Axio Vert. A1, Carl Zeiss, Oberkochen, Germany) and estimating the number of nucleated cells per 10 million platelets in a Neubauer counting chamber (Cat. no. 717805, BRAND GMBH + CO KG, Wertheim, Gemany) (Supplementary Fig 4a–c; ZEN microsopy software 3.11, Carl Zeiss, Oberkochen, Germany). For flow cytometry analyses, purity was controlled by additional gating on leukocyte markers (see Supplementary Fig. 4d).

## Proteomics sample preparation
Proteins (10 μg) were run for 0.5 cm into an SDS-PAGE and the entire piece was cut out and digested using trypsin according to Shevchenko et al.[36] adapted for the DigestPro MSi robotic system (INTAVIS Bioanalytical Instruments AG, Tuebingen, Germany).

## Flow-cytometry analysis
Shortly prior to analysis, samples were taken out of −80 °C storage and briefly thawed in a waterbath (37 °C). They were then resuspended in 1 ml of wash buffer (WB; Tyrode's buffer (cat. no T2397-1L, Merck KGaA, Darmstadt, Germany) + 100 mg BSA (cat. no. 9048-46-8, Carl Roth GmbH + Co. KG, Karlsruhe, Germany)) while working carefully to avoiding bubbles and mechanical stress. Cells were spun at 400 G for 5 min at RT. After straining through a 70uM filter, they were then stained with an antibody cocktail (see Supplementary Table 2) for 20 min at RT in the dark. After an additional wash step with WB, platelets were fixed using the cytofix/cytoperm kit (cat. No. 554714, BD Biosciences, California, USA) and additionally stained with antibodies coupled to intracellular markers (see Supplementary Table 2). All reagents were tested and titrated prior to use in experiments. Titrations were repeated for new commercial lots where appropriate. All antibody incubations were carried out in Tyrode's buffer supplemented with BSA and 20% Brilliant Stain Buffer (cat. no. 563794, BD Biosciences, California, USA). The samples were washed once in Perm/Wash buffer (eBioscience, California, USA) and centrifuged to pellet the cells before resuspension in Tyrode's Buffer prior to flow-cytometry acquisition. Samples were acquired the same day, on aCytek Aurora spectral analyser (Cytek Biosciences, California, USA) using SpectroFlo Software (version 2.0) following daily quality control procedures as instructed by the manufacturer.

## MS method Orbitrap Exploris 480
A liquid chromatography- tandem mass spectrometry (LC-MS/MS) analysis was carried out on an Ultimate 3000 Ultra-High Performance Liquid Chromatography (UPLC) system (Thermo Fisher Scientific)

directly connected to an Orbitrap Exploris 480 mass spectrometer for a total of 120 min. Peptides were online desalted on a trapping cartridge (cat. no. 174500; Acclaim PepMap300 C18, 5 μm, 300 Å wide pore; Thermo Fisher Scientific) for 3 min using 30 μl/min flow of 0.05%v/v trifluoroacetic acid (TFA; cat. no. 00202341A8BS; Biosolve) in water. The analytical multistep gradient (300 nl/min) was performed using a nanoEase MZ Peptide analytical column (cat. no. 186008794; 300 Å, 1.7 μm, 75 μm x 200 mm, Waters) using solvent A (0.1%v/v formic acid in water; cat. no. 00069141A8BS, Biosolve and cat. no. W6-4, Thermo Fisher Scientific) and solvent B (0.1%v/v formic acid in acetonitrile; cat. no. A955-212; Thermo Fisher Scientific). For 102 min the concentration of B was linearly ramped from 4% to 30%, followed by a quick ramp to 78%. After two minutes the concentration of B was lowered to 2% and a 10 min equilibration step appended. Eluting peptides were analyzed in the mass spectrometer using data-independent acquisition (DIA) mode. A full scan at 120k resolution (380-1400 m/z, 300% automatic gain control (AGC) target, 45 ms maximum injection time (maxIT)) was followed by 47 MS2 (DIA) windows covering the mass range from 400 to 1000 m/z with variable width (30k resolution, AGC target 1000%, maxIT 54 ms, with 1 m/z overlap, 28% higher-energy collisional dissociation (HCD) collision energy). System performance was constantly monitored via an internal tool using regular (approx. every two days) injections of a quality control (QC) sample.

## Data analysis, spectral flow cytometry
From the raw data acquired by spectral flow-cytometry, using FlowJo software version 10.6.2 and 10.7.1 (TreeStar, BD, Oregon, USA), live platelets were identified using manual gating on FSC versus SSC. For flow-cytometry data, the compensation matrix was corrected in FlowJo (TreeStar, BD, Oregon, USA) by pre-gating on live platelets, and the total fraction of live, singlet platelets was exported. Data were then transformed with an inverse hyperbolic sine (arcsinh) function (cofactors ranging between 5 and 18000) and imported into the R environment (version 3.6.1) for subsequent analysis[37].

The high dimensional analysis was carried out using the R environment, based on the workflow described previously by Hartmann et al.[38] Briefly, UMAPs were generated using the package *umap* version 0.2.7.0[12], and FlowSOM clustering was overlaid on the dimensionality reduction maps[13]. Frequency plots were generated using the *ggplot2* package version 3.3.5, and heatmaps were generated using the *pheatmap* package version 1.0.12.

Principal Component Analyses (PCA) were carried out using the PCATest R package. Statistical significance testing of the principal components was performed using permutation-based methods within PCATest (Supplementary Table 3).

## Data analysis, proteomics
Analysis of DIA RAW files was performed with Spectronaut (Biognosys, version 17.1.221229.55965) in directDIA+ (deep) library-free mode. Default settings were applied with the following adaptions. Within the Pulsar Search in Result Filters the m/z Max was set to 1800 and Min to 300, the Relative Intensity was set to 5%. Within DIA Analysis under

Quantification the Protein LFQ Method was set to MaxLFQ. The data was searched against the human proteome from Uniprot (human canonical reference database, containing 81,837 unique entries from 26.10.2022).

Downstream statistical analysis was then performed in R (version 4.2.2). For this, the Spectronaut output was imported into R using the package protti (version 0.6.0). Imported data was then further processed using the R package proteus (version 0.2.16). As part of the analysis with proteus, data was log2 transformed, followed by the Exploratory Data Analysis which included visualization of sample grouping (PCA, Hierarchical clustering); protein expression (Heatmaps) and protein expression sets and their intersections (Upset Plots). Although proteus offers wrappers around Limma functions, differential expression (DE) analysis was performed with the R package Limma (version 3.54.1) outside proteus to have more flexibility in linear modeling (see Supplementary Table 4–7). For graphical visualization of DE expression analysis results volcano plots were produced in R to show statistical significance -log10(adj. P. Val) versus log2 FC, and log10 (adj. P. Val) versus mean expression of total platelet proteins (see Supplementary Fig. 2–3).

Proteomics difference in skin diseases: In order to identify differentially expressed proteins between the skin diseases at the different post therapy timepoints and the baseline (prior to therapy) and the healthy volunteers, respectively, a simple linear model was fitted using Limma to each protein consisting of a fixed effect for a combined factor of the two main experimental factors skin disease (factor levels: Pso (psoriasis) and AD (atopic dermatitis)) and therapy time (factor levels: baseline=prior to therapy, 1 month = 1 month post systemic therapy, 4 months = 4 months post systemic therapy initiation). Using this approach separate coefficient for each of these factor combinations is built in the linear model fit, which then allows to extract comparisons of interest (e.g. Pso-4month vs Healthy controls) as contrasts. The final results include statistics for each protein including empirical Bayes moderated nominal $p$ values which were adjusted for multiple testing by controlling the false discovery rate (FDR) using the Benjamini–Hochberg procedure[39]. As threshold, a protein was called DE with a multiple adjusted p ≤ 0.05%. No log fold change filter criterion was applied for statistical assessment to find DE proteins.

## Statistics

Descriptive statistics as frequency, mean, range and standard deviation were used to analyse the human cohort. Kolmogorov-Smirnov test was used to investigate normality of the data. Where appropriate, Spearman correlations were applied. Multiple linear models were fitted for confounders and residuals visualized the using boxplots. Statistical analysis was carried out using the R package rstatix v.0.7.0. Briefly, unpaired t tests provided p values, which were then adjusted for multiple comparisons using a Benjamini–Hochberg (BH) test. P values of less than 0.05 were considered significant and are indicated by an asterisk (*) or the numerical value on the respective graphs[39].

## Ethics

This study was conducted according to the Declaration of Helsinki. Patients and donors provided written and informed consent, and the study was approved by the ethics committee of Heidelberg (approval no. S-834/2020).

## Reporting summary

Further information on research design is available in the Nature Portfolio Reporting Summary linked to this article.

## Data availability

The FCS files of the spectral flow-cytometry data are available for download through the following repository link: https://doi.org/10.5281/zenodo.14052849. The proteomics data generated in this study have been deposited in via PRIDE (http://www.ebi.ac.uk/pride; Project accession: PXD057615) [https://www.ebi.ac.uk/pride/archive/projects/PXD057615]. The raw patient specific data are not available due to data privacy laws. All other data are available in the article and its Supplementary files or from the corresponding author upon request. Source data are provided with this paper.

## Code availability

An R-based script for data analysis is available through the following Repository link: https://doi.org/10.5281/zenodo.14052849.

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

## Acknowledgements

We would like to thank Kornelia Junge, Olivia Bochnig, Therezia Bokor-Billmann, Silvia Mihalceanu, and Franziska Keidel for their support in patient coordination. We further thank the team of the Proteomcis Core Facility for help with sample preparation and LC-MS/MS analysis. KSK is funded by the Physician-Scientist Program of Heidelberg University, Faculty of Medicine. TB received funding from the Federal Ministry of Education and Research (BMBF) and the Ministry of Science Baden-Württemberg within the framework of the Excellence Strategy of the Federal and State Governments of Germany. This work was further supported by TRR156/2–246807620 ("The Skin as Sensor and Effector Organ Orchestrating Local and Systemic Immune Responses", DFG; KSK, AE and MH); the Cluster of Excellence iFIT EXC 2180 ("Image-Guided and Functionally Instructed Tumor Therapies", University of Tübingen, Tübingen, Germany; (MH)); The skintegrity network the Swiss National Science Foundation (310030_188450, and 291 310030_219287; BB) and the European Research Council (ERC; 882424; BB).

## Author contributions

All authors contributed to the study conception and design. Patient recruitment by K.S.K., K.S., A.E.. Supervision of method establishment by B.B. and M.H.. Experimental design by K.S.K., S.Kr., B.B., and M.H.. Sample isolation by K.S.K., P.R., S.Ka., T.B. Sample processing and flow cytometry experiments performed by S.Kr., E.F., L.R., and P.H.. Data analysis and visualization by S.Kr. and K.S.K.. Support with experiments by S.P., F.M., U.R., and Y.W.. Proteomics analysis by F.B., S.C., S.N., D.H., M.S.. Interpretation of the data by all authors. The first draft of the manuscript was written by K.S.K., S.Kr., B.B., and M.H.. Both K.S.K. and S.Kr. contributed equally and have the right to list their name first in their C.V.. Additionally, T.B. and E.F. contributed equally to this work. All authors commented on previous versions of the manuscript. All authors read and approved the final manuscript.

## Funding

## Competing interests

K.S.K. discloses to have received travel grants by Almirall, Janssen and Eli Lilly. K.S. discloses that he was an advisor and/or received speakers' honoraria and/or received grants and/or participated in clinical trials of the following companies: AbbVie, Almirall, Boehringer Ingelheim, Celgene, Eli Lilly, Galderma, Janssen-Cilag GmbH, LEO Pharma, Novartis, Pfizer and UCB. A.E. discloses speakers' honoraria/travel grants for Janssen-Cilag GmbH, Abbvie, and Eli Lilly. The authors declare that these conflicts are not associated with this work. The other authors have no competing interests to declare.

## Additional information

**Peer review information** : *Nature Communications* thanks Laura Calabrese, Christian Sadik and the other, anonymous, reviewer(s) for their contribution to the peer review of this work. A peer review file is available.

[1]Division of Chronic Inflammation and Cancer, German Cancer Research Center (DKFZ), Heidelberg, Germany. [2]Department of Dermatology, University Hospital Heidelberg, Heidelberg, Germany. [3]Institute of Experimental Immunology, University of Zurich, 8057 Zurich, Switzerland. [4]Interdisciplinary Center for Scientific Computing (IWR), Heidelberg University, Heidelberg, Germany. [5]Institute for Immunology, University Hospital Heidelberg, Heidelberg, Germany. [6]Quantitative Biology Center (QBiC), University of Tübingen, Tübingen, Germany. [7]The M3 Research Centre, Medical faculty Tübingen, Tübingen, Germany. [8]Institute for Bioinformatics and Medical Informatics, University of Tübingen, Tübingen, Germany. [9]Cluster of Excellence iFIT (EXC 2180) "Image-Guided and Functionally Instructed Tumor Therapies", Eberhard Karls University, Tübingen, Germany. [10]Proteomics Core Facility, German Cancer Research Center (DKFZ), Heidelberg, Germany. [11]Institute for Interdisciplinary Research on Cancer Metabolism and Chronic Inflammation, The M3 Research Centre, Medical faculty Tübingen, Tübingen, Germany. [12]Present address: Roche Pharma Research and Early Development, Strategy, Portfolio and Operations, Roche Innovation Center Basel, F.Hoffmann-La Roche Ltd, Grenzacherstrasse 124, Basel, Switzerland. [13]Present address: Department of Neuropathology, Charité - Universitätsmedizin Berlin, Berlin, Germany. [14]These authors contributed equally: Katharina S. Kommoss, Sinduya Krishnarajah. [15]These authors jointly supervised this work: Burkhard Becher, Mathias Heikenwälder. ✉e-mail: becher@immunology.uzh.ch; mathias.heikenwaelder@med.uni-tuebingen.de

