## [Transparent Peer Review file · Nature Communications]

Single cell mapping identifies a distinct platelet-phenotype in psoriatic Type III inflammation

Corresponding Author: Dr Mathias Heikenwalder

Version 0:

Reviewer comments:

Reviewer #1

(Remarks to the Author)

Kommos et al. report the finding of unique subpopulations of platelets with an activated and proinflammatory phenotype in patients with psoriasis but not atopic dermatitis. This finding is significant and potentially of great importance for the pathogenesis of psoriasis. The put great effort and expertise into the state-of-the-art proteomic analysis of platelets. The basic concept of the paper is innovative: platelets have not been investigated in great detail in inflammatory diseases of the skin although they would be easily accessible as biomarkers. The work generates the basis to potentially distinguish psoriasis from other inflammatory skin diseases and to predict flares and treatment responses based on the phenotype of platelets in the future.

Reviewer #2

(Remarks to the Author)

In this manuscript, the authors present interesting findings that underscore differences in platelet phenotypes among chronic inflammatory skin diseases—psoriasis (type III inflammation), atopic dermatitis (type II inflammation), and healthy donors. While platelet heterogeneity is an underexplored aspect of chronic inflammatory diseases, the study is largely descriptive and does not delve into mechanistic insights. Moreover, it remains unclear whether the observed variability in platelet phenotypes is specific to type III inflammation in the skin or represents a broader characteristic of type III/I inflammation in general.

A notable limitation is the small number of clinical participants and the aggregation of data from patients receiving diverse biological treatments. These treatments target distinct components of the inflammatory cascade and may differently influence platelet behavior before the resolution of pathogenic skin alterations. Additionally, the identification of a distinct platelet phenotype specific to psoriasis/type III inflammation relies on sophisticated methodological approaches, such as high-dimensional flow cytometry.

For these reasons, the potential diagnostic applicability of these findings is currently limited.

Other concerns/suggestions;

Correct Line 90-91 to include Th17 cells: "e.g., T-helper (Th1)/Th2/Th17 instead of Th1)/Th2 cells and their associated cytokines (e.g., IFN- γ , IL-4, IL-17)."

Include information about the purity of platelet preparations and the functional assays performed on platelets from different donor groups.

Reviewer #3

(Remarks to the Author)

Dear Authors,

This is an interesting and well-articulated article presenting novel findings on a distinct platelet phenotype in psoriatic type III inflammation.

I have a few minor suggestions for improvement:

- 1) The Introduction section could be streamlined by reducing the background information on inflammation types, as this is likely familiar to the target audience. Instead, I recommend emphasizing the novelty and significance of platelet phenotyping in inflammatory contexts.
- 2) In the paragraph "MS method Orbitrap Exploris 480", abbreviations should be explicitly defined for clarity.
- 3) On line 314, the authors mention a trend for a positive correlation between CD32+CD154+ frequencies and PASI, reflecting the clinical extent of skin involvement. However, PASI measures not only the extent of skin involvement but also the severity of lesions. This sentence should be revised accordingly.
- 4) On line 269, the authors state that the clinical presentation of skin inflammation in psoriasis and AD was comparable based on PASI and EASI scores. However, these tools assess distinct features of two different diseases. Therefore, it is not entirely accurate to claim that the clinical presentations were comparable.
- 5) Certain comparisons, such as those between psoriasis patients and healthy controls, could benefit from adjustments for potential confounding factors, such as BMI.
- 6) Including a paragraph discussing the potential therapeutic applications of platelet profiling would provide a valuable forward-looking perspective and enhance the clinical relevance of the findings.

Version 1:

Reviewer comments:

Reviewer #2

(Remarks to the Author)

I found the studies much improved, particularly with the inclusion of additional patient cohorts and detailed information on platelet purity. While the paper presents novel and interesting findings by highlighting inflammation-associated "platelet phenotypes," it remains purely descriptive and currently offers limited diagnostic value.

The manuscript does not sufficiently address mechanistic insights, nor does it convincingly demonstrate the specificity of the distinct inflammatory platelet signature to psoriasis, as similar signatures may be observed in other Th1/Th17-driven disorders.

Furthermore, skin inflammation-related changes in platelets have already been reported in psoriasis patients, including elevated mean platelet volume (MPV) and platelet count. Existing markers used to assess resolving inflammation may provide comparable diagnostic value without the need for the proposed, more complex approach.

Without additional mechanistic insights or evidence of disease specificity, the studies remain of limited value.

Minor point:

Consider changing the order of T-helper cell subsets and their associated cytokines to:

T-helper 1 (Th1)/Th17/Th2 cells and their cytokines (e.g., IFN- γ , IL-17, IL-4), as IL-4 is more closely associated with Th2 responses, while IL-17 is characteristic of Th17 cells.

Reviewer #3

(Remarks to the Author)

Dear Authors,

You have carefully considered each of my suggestions and made thoughtful and appropriate revisions throughout the manuscript. The additional analyses related to BMI and the inclusion of a forward-looking discussion on the therapeutic potential of platelet profiling are especially valuable and enhance the overall impact of the work.

Thank you for your thorough revisions.

Reviewer #1:

Kommos et al. report the finding of unique subpopulations of platelets with an activated and proinflammatory phenotype in patients with psoriasis but not atopic dermatitis. This finding is significant and potentially of great importance for the pathogenesis of psoriasis. The put great effort and expertise into the state-of-the-art proteomic analysis of platelets. The basic concept of the paper is innovative: platelets have not been investigated in great detail in inflammatory diseases of the skin although they would be easily accessible as biomarkers. The work generates the basis to potentially distinguish psoriasis from other inflammatory skin diseases and to predict flares and treatment responses based on the phenotype of platelets in the future.

Response:

We thank Reviewer #1 for the thoughtful comments and their recognition of the strengths of our methodology and conclusions. We share their interest in elucidating distinct inflammatory signatures in dermatological diseases and are committed to advancing this line of investigation. The findings presented in this manuscript lay an important foundation for our ongoing work in this area.

Reviewer #2:

In this manuscript, the authors present interesting findings that underscore differences in platelet phenotypes among chronic inflammatory skin diseases—psoriasis (type III inflammation), atopic dermatitis (type II inflammation), and healthy donors.

Comment 1:

While platelet heterogeneity is an underexplored aspect of chronic inflammatory diseases, the study is largely descriptive and does not delve into mechanistic insights. Moreover, it remains unclear whether the observed variability in platelet phenotypes is specific to type III inflammation in the skin or represents a broader characteristic of type III/I inflammation in general.

Response 1:

We thank Reviewer #2 for their insightful comments. Within this study, we wanted to specifically characterize platelets prospectively in an unbiased manner to gain insights into the platelet phenotype under type II/III inflammation. Unravelling the mechanistic underpinnings of the contribution of platelets to this inflammation was beyond the scope of this study. Several hypotheses have been proposed regarding the pathophysiological role of platelets in psoriasis and atopic dermatitis. Unlike the *de novo* synthesis of TNF- α by platelets - which functions in both its transmembrane and soluble forms - the inflammatory signaling of CD154 (a member of the TNF superfamily) is more rapid and is only activated upon ligation with CD40 on endothelial cells or monocytes.¹ Hence, CD154 may act as a signal-amplifier within psoriatic type III inflammation, which is primarily driven by inflammatory leukocytes. Additionally, CD32 upregulation has been shown to be regulated by interferon gamma,² which may explain its increased expression in individuals with psoriasis. In sepsis, platelet TLR4 has been shown to activate neutrophils and promote the formation of neutrophil extracellular traps (NET). Given that NET-formation is a hallmark of psoriasis,³ TLR4 on platelets may similarly contribute to this self-amplifying inflammation. TLR2 activation was shown to lead to platelet release of sCD62p, RANTES and sCD40-ligand,⁴ promoting a feed-forward loop of type III inflammation. We have added a paragraph on these potential mechanistic links in the discussion (pp. 14-15, II. 395-415).

Additionally, we fully agree that additional non-dermatological studies investigating Type II/III diseases (i.e. allergic asthma; chronic inflammatory bowel disease), and even extension to type I inflammation (i.e. lupus erythematoses) are warranted and needed. To add these diseases in a prospective manner was unfeasible for us, but we have included a statement in

the limitations highlighting the need for this (p. 16, ll. 442-445). We hope that the findings in this manuscript lay basis for the initiation of platelet phenotype investigations in a broader inflammatory context.

Comment 2:

A notable limitation is the small number of clinical participants and the aggregation of data from patients receiving diverse biological treatments. These treatments target distinct components of the inflammatory cascade and may differently influence platelet behavior before the resolution of pathogenic skin alterations. Additionally, the identification of a distinct platelet phenotype specific to psoriasis/type III inflammation relies on sophisticated methodological approaches, such as high-dimensional flow cytometry.

For these reasons, the potential diagnostic applicability of these findings is currently limited.

Response 2:

We thank the Reviewer #2 for their valid comment regarding the sample size. Indeed, within the submitted manuscript we reported on n=27 psoriasis patients, n=4 AD patients and n=10 controls in the first cohort, and n=6 psoriasis patients, n=10 AD patients and n=6 controls in the validation cohort 2.

To further validate the findings, we have added an additional third cohort consisting of n=21 psoriasis patients, n=9 AD patients and n=8 controls (**Suppl. Fig. 1f**). Overall, this study therefore investigated n=54 psoriasis patients, n=23 AD patients and n=24 controls (reflected in updated schematic overview of the study, **Fig. 1a**).

The processing and measurement of the third cohort was only possible with the help of Lukas Rindlisbacher and Philipp Häne, which is reflected in their additional authorship.

The results from our third retrospective cohort closely mirror those reported in the earlier version of our manuscript, further reinforcing the observed trends through additional flow cytometric analysis. Specifically, we present data demonstrating that the identified platelet subtypes are enriched in psoriatic patients at Baseline, with significantly higher frequencies compared to healthy volunteers and patients with atopic dermatitis. The inflammatory subset, marked by CD32⁺CD154⁺ expression, progressively declines during treatment and resolves with clinical improvement. Additionally, the expression patterns of TLR4⁺TLR2⁺ platelets observed in our initial experimental cohorts are consistently replicated in this independent dataset.

Point by point, Figure 1: Frequency of indicated platelet clusters in psoriasis patients of cohort three over time. Samples represent psoriasis patient platelets prior to (baseline), one month (T1) and four months (T4) after start of systemic therapy. Data was analyzed with an unpaired student's t-test.

The reviewer rightly highlights the diversity of biological treatments received by patients in this study, ranging from specific cytokine-blocking biologics to broader anti-inflammatory agents such as methotrexate and JAK-inhibitors. However, the sample size within each treatment group is too small to support meaningful sub-group analyses or conclusions about the ability of the observed platelet phenotype to predict therapeutic response. Notably, the consistent effect of these diverse treatments on the same platelet phenotype cluster suggests that the observed resolution of platelet subclusters reflects overall disease resolution, rather than a treatment-specific effect.

We agree with this Reviewer that the applying these findings in routine clinical practice is currently limited due to the time-, cost- and specialized expertise required for the methods used. However, this study lay the basis for further investigation into platelets in diagnostics and therapy in inflammatory diseases.

Comment 3:

Correct Line 90-91 to include Th17 cells: "e.g., T-helper (Th1)/Th2/Th17 instead of Th1)/Th2 cells and their associated cytokines (e.g., IFN- γ , IL-4, IL-17)."

Response 3:

We have included Th17 cells in line 106, page 5.

Comment 4:

Include information about the purity of platelet preparations and the functional assays performed on platelets from different donor groups.

Response 4:

We thank the Reviewer for this important comment. We have added respective passages to the platelet isolation methods (p. 5, ll. 144-147; p.6, l.155 et seqq.) and addressed this topic in a **new Supplemental Methods Figure 1**.

The applied protocol for platelet isolation by Best et al., 2019, was developed for platelet RNA sequencing. They described a highly purified platelet preparation.⁵ We randomly performed control of platelet purity by microscopy and estimating the number of nucleated cells per 10 million platelets in a Neubauer counting chamber, and have added representative images of this in **Suppl. Methods Fig. 1a-c**.

To further strengthen Reviewer #2's confidence in our platelet purity, we have added our flow cytometry gating strategy to the manuscript, which is specifically designed to exclude contaminating immune cells and selectively identify bona fide platelets using conventionally accepted lineage markers. Exemplary frequencies are demonstrated for cohort 3, which was requested within the current revision process, in **Point by Point Table 1**. As is evident with the exemplary gating and frequency of Non-CD45-Non-CD3, Non-CD19 positive cells, immune cell contamination was not an issue that was encountered in the cohorts acquired for this study. The exemplary gating strategy, which was applied to all three investigated cohort, has been added to the manuscript in **Suppl. Methods Fig. 1d**.

Sample Number	True Platelets (%)	Non-CD45-Non-CD3 positive cells (%)	Non-CD19 positive cells (%)
1	92.46	99.44	99.86
2	77.23	99.00	99.96
3	98.66	99.90	99.99
4	80.85	99.19	99.81
5	92.81	98.88	99.82
6	87.82	99.24	99.84
7	65.64	99.44	99.96
8	86.01	99.09	99.97
9	87.04	98.95	99.88
10	96.25	99.05	99.98
11	90.39	98.85	99.99
12	74.59	99.34	99.97
13	74.51	99.23	99.99
14	73.76	99.02	99.97
15	70.67	99.35	99.99
16	95.35	98.31	99.92
17	98.76	98.79	99.80
18	97.24	98.38	99.97
19	98.51	99.26	99.96
20	98.56	99.70	99.91
21	89.36	98.78	99.93
22	93.04	99.15	99.99
23	90.01	99.41	99.96
24	82.19	99.45	99.80
25	96.39	97.68	99.77
26	98.29	99.67	99.91
27	92.61	99.85	99.98
28	97.72	97.49	99.73
29	96.90	99.06	99.86
30	90.86	98.00	99.96
31	97.50	99.75	99.94
32	99.11	99.79	99.95
33	98.39	99.70	99.80
34	81.78	98.56	99.99
35	97.48	99.81	99.98
36	97.55	99.73	99.93
37	97.28	99.57	99.93
38	96.37	99.85	99.98
39	92.88	99.22	100.00
40	97.13	99.75	99.99
41	98.29	99.74	99.98
42	89.41	99.66	99.98
43	97.51	99.72	99.97
44	96.81	99.68	99.99
45	85.74	97.95	100.00
46	96.85	98.92	99.83
47	93.05	99.52	99.93
48	98.14	99.56	99.95
49	98.57	99.61	99.96
50	97.07	99.72	99.99
51	98.99	99.64	99.98
52	96.72	99.82	99.96

53	98.89	99.60	99.91
54	76.62	99.11	99.98
55	96.56	99.64	99.86
56	96.67	99.30	99.96
57	98.34	99.78	99.96
58	98.54	99.59	99.92
59	98.40	99.71	99.86
60	97.35	99.55	99.87
61	97.64	99.79	99.97
62	98.40	99.69	99.96
63	96.63	99.34	99.80
64	98.43	99.48	99.89

Point by Point Table 1: Investigated samples are highly platelet purified. Frequencies of indicated subgroups in % for cohort three, based on the gating strategy in **Supplemental Material Figure 1**. Gating on “true platelets” was achieved after gating on single cells using forward-/side scatters.

New Supplemental Methods Figure 1: Platelet samples contain minor leukocyte contamination. **a-c.** Representative images of three individual patient platelet pellets of 9 ml of blood samples used for proteomic analyses. The platelet pellet was resuspended in 50 μ l of Tyrode's buffer, and 1 μ l of the suspension diluted 1:50 in Tyrode's buffer. Pictures were taken at original image x 100 magnification within a Neubauer chamber. Arrows indicate leukocytes. **d.** Gating strategy of high dimensional flow cytometry analysis with representative plots.

We agree with this Reviewer that functional assays on the platelets of the individual subgroups are of high interest. However, as described above, this was out of the scope of the current manuscript. Based on the findings of this study, assays for platelet attraction and activation in a controlled type III/II environment are necessary and aimed to be deciphered in future studies.

Reviewer #3:

This is an interesting and well-articulated article presenting novel findings on a distinct platelet phenotype in psoriatic type III inflammation.

I have a few minor suggestions for improvement:

Comment 1:

The Introduction section could be streamlined by reducing the background information on inflammation types, as this is likely familiar to the target audience. Instead, I recommend emphasizing the novelty and significance of platelet phenotyping in inflammatory contexts.

Response 1:

We thank this Reviewer for their constructive comments. We have rewritten the introduction to lay emphasis on platelets and the novelty and significance of platelet phenotyping in inflammatory contexts and minimized general background on T-cell based inflammation types, as the reviewer suggested (p. 5, ll. 89-108). This restructure of the introduction now better reflects platelets as the key of the manuscript, and we kindly thank the reviewer for this important remark.

Comment 2:

In the paragraph "MS method Orbitrap Exploris 480", abbreviations should be explicitly defined for clarity.

Response 2:

We thank this Reviewer for this important remark and have revised the "MS method Orbitrap Exploris 480" paragraph to clarify all abbreviations used (p. 7, l. 184 et seqq.).

Comment 3:

On line 314, the authors mention a trend for a positive correlation between CD32+CD154+ frequencies and PASI, reflecting the clinical extent of skin involvement. However, PASI measures not only the extent of skin involvement but also the severity of lesions. This sentence should be revised accordingly.

Response 3:

We have corrected the above-mentioned sentence to correctly reflect the meaning of the psoriasis area and severity score (p. 12, l. 332), as the reviewer kindly pointed out.

Comment 4:

On line 269, the authors state that the clinical presentation of skin inflammation in psoriasis and AD was comparable based on PASI and EASI scores. However, these tools assess distinct features of two different diseases. Therefore, it is not entirely accurate to claim that the clinical presentations were comparable.

Response 4:

We thank this Reviewer for this important remark and have rewritten the sentences comparing PASI and EASI to more adequately reflect the meaning behind the clinical scores (p. 11, ll. 280-285).

Comment 5:

Certain comparisons, such as those between psoriasis patients and healthy controls, could benefit from adjustments for potential confounding factors, such as BMI.

Response 5:

We thank this Reviewer for this valuable comment.

In line with previous reports, psoriasis patients in our study had a significantly higher body mass index (BMI) compared to healthy controls and individuals with atopic dermatitis (AD). Importantly, over the four-month study period, BMI remained largely stable within individual patients. Since we observed a consistent intra-individual decrease in the CD32⁺CD154⁺ platelet cluster in psoriasis patients over time, despite no significant BMI changes, this suggests that the observed phenotype is not BMI-driven (see also p. 14, ll. 374-379).

To further explore whether the baseline frequency of the CD32⁺CD154⁺ cluster is influenced by BMI or more strongly associated with disease group, we conducted a Spearman correlation analysis between BMI and CD32⁺CD154⁺ frequency across the entire cohort and within each group. While a significant positive correlation was found in the overall cohort (Spearman's rho = 0.56, p = 0.00016), no significant correlations were observed within individual groups: Control (rho = 0.09, p = 0.808), AD (rho = 0.40, p = 0.75), and Psoriasis (rho = 0.16, p = 0.435).

To assess whether group-specific effects persist independently of BMI, we fitted a linear model (baseline CD32⁺CD154⁺ frequency ~ BMI) and analyzed the residuals using boxplots stratified by disease group. This residual analysis confirmed statistically significant differences between groups.

Finally, we repeated the analysis in a BMI-matched subgroup (BMI < 25), which again revealed significantly higher CD32⁺CD154⁺ frequencies in psoriasis patients compared to controls and AD patients.

Together, these results suggest that elevated CD32⁺CD154⁺ frequencies are more strongly associated with disease group than with BMI. We have now added these findings to page 12, ll. 325-328, and included them **in Supplementary Figure 1a–c**.

Comment 6:

Including a paragraph discussing the potential therapeutic applications of platelet profiling would provide a valuable forward-looking perspective and enhance the clinical relevance of the findings.

Response 6:

We thank the reviewer for this critical remark and have introduced a paragraph into the discussion highlighting the potential diagnostic and therapeutic use of platelet profiling in inflammatory diseases (p. 15, l. 422 et seq.). Further, we have edited the final sentence of the conclusion to highlight this forward-looking perspective accordingly (p. 16, ll. 448-450).

REFERENCES

- 1 Henn, V., Steinbach, S., Buchner, K., Presek, P. & Kroczeck, R. A. The inflammatory action of CD40 ligand (CD154) expressed on activated human platelets is temporally limited by coexpressed CD40. *Blood* **98**, 1047-1054 (2001). <https://doi.org/10.1182/blood.v98.4.1047>
- 2 Schneider, D. J. & Taatjes-Sommer, H. S. Augmentation of megakaryocyte expression of FcγRIIIa by interferon gamma. *Arterioscler Thromb Vasc Biol* **29**, 1138-1143 (2009). <https://doi.org/10.1161/ATVBAHA.109.187567>
- 3 Herster, F. *et al.* Neutrophil extracellular trap-associated RNA and LL37 enable self-amplifying inflammation in psoriasis. *Nat Commun* **11**, 105 (2020). <https://doi.org/10.1038/s41467-019-13756-4>
- 4 Damien, P. *et al.* NF-κB Links TLR2 and PAR1 to Soluble Immunomodulator Factor Secretion in Human Platelets. *Front Immunol* **8**, 85 (2017). <https://doi.org/10.3389/fimmu.2017.00085>
- 5 Best, M. G., In 't Veld, S., Sol, N. & Wurdinger, T. RNA sequencing and swarm intelligence-enhanced classification algorithm development for blood-based disease diagnostics using spliced blood platelet RNA. *Nat Protoc* **14**, 1206-1234 (2019). <https://doi.org/10.1038/s41596-019-0139-5>

Reviewer #2:

I found the studies much improved, particularly with the inclusion of additional patient cohorts and detailed information on platelet purity. While the paper presents novel and interesting findings by highlighting inflammation-associated “platelet phenotypes,” it remains purely descriptive and currently offers limited diagnostic value.

The manuscript does not sufficiently address mechanistic insights, nor does it convincingly demonstrate the specificity of the distinct inflammatory platelet signature to psoriasis, as similar signatures may be observed in other Th1/Th17-driven disorders.

Furthermore, skin inflammation-related changes in platelets have already been reported in psoriasis patients, including elevated mean platelet volume (MPV) and platelet count. Existing markers used to assess resolving inflammation may provide comparable diagnostic value without the need for the proposed, more complex approach.

Without additional mechanistic insights or evidence of disease specificity, the studies remain of limited value.

Response #2:

We thank Reviewer #2 for their thoughtful evaluation and for recognizing the improvements in the revised manuscript, particularly the inclusion of expanded patient cohorts and the more detailed reporting on platelet purity.

We appreciate the concern regarding the absence of mechanistic insights, as this was beyond the scope of the present study. However, we respectfully maintain that our findings provide meaningful and novel contributions. The characterization of inflammation-associated platelet phenotypes in psoriasis, especially within well-defined human cohorts, lays an important foundation for understanding the complex interplay of platelets across different inflammatory contexts. We hope that the data presented here will stimulate future investigations into the potential mechanistic links discussed in both the literature and our manuscript (p. 8, l. 207 et seq.).

As noted in the limitations of the study (p. 10, ll. 254–257), we fully agree that investigations in additional Th1/Th17- and Type I/II-driven diseases will be required to confirm the specificity of the observed platelet phenotype. In our view, this study provides a valuable platform upon which future research can build.

We also acknowledge that changes in platelet indices such as mean platelet volume and platelet count have already been reported in psoriasis. However, these previous findings lack the single-cell granularity afforded by the high-dimensional flow cytometry approach employed in our work. This distinction is underscored by our bulk proteomics data, which did not detect the platelet subpopulations revealed through single-cell analysis. We believe that this higher-resolution approach can uncover patterns not captured by bulk measurements and may ultimately provide complementary, or even superior, insights once validated in larger and more mechanistic studies.

While established markers of resolving psoriatic inflammation may indeed yield comparable diagnostic value, platelet phenotyping offers potential advantages when compared to other methods used for diagnostic distinction in clinically challenging dermatological inflammatory diseases (e.g. single-cell RNA sequencing¹, spatial transcriptomics², bulk NanoString RNA sequencing³). Platelet phenotyping appears to be less invasive and potentially more efficient (see discussion, p. 9, ll. 234-238). Importantly, it may also hold future therapeutic relevance, as noted in the discussion (p. 9, l. 239 et seq.). Therapeutic interventions require precise targets in the context of platelets, given that platelet-specific treatments carry the inherent risk

of bleeding complications. The data presented here provide the first psoriasis-specific platelet markers that may serve as a foundation for such advances.

Comment 1:

Consider changing the order of T-helper cell subsets and their associated cytokines to: T-helper 1 (Th1)/Th17/Th2 cells and their cytokines (e.g., IFN- γ , IL-17, IL-4), as IL-4 is more closely associated with Th2 responses, while IL-17 is characteristic of Th17 cells.

Response 1:

We thank Reviewer #2 for indication of the confusing order of helper cells and associated cytokines. We have revised the text accordingly (p. 4, ll. 74-76).

Reviewer #3:

Dear Authors,

You have carefully considered each of my suggestions and made thoughtful and appropriate revisions throughout the manuscript. The additional analyses related to BMI and the inclusion of a forward-looking discussion on the therapeutic potential of platelet profiling are especially valuable and enhance the overall impact of the work.

Thank you for your thorough revisions.

Response 1:

We sincerely thank Reviewer #3 for their careful evaluation of our manuscript and their insightful suggestions, which have significantly strengthened and refined our work. In particular, we are grateful for their input that prompted the inclusion of BMI-corrected analyses and the expanded discussion on the therapeutic potential of platelet profiling, both of which enhance the overall impact of the manuscript.

REFERENCES

- 1 Chennareddy, S. et al. Single-cell RNA sequencing of chronic idiopathic erythroderma defines disease-specific markers. *J Allergy Clin Immunol* 155, 892-908 (2025). <https://doi.org/10.1016/j.jaci.2024.11.037>
- 2 Schabitz, A. et al. Spatial transcriptomics landscape of lesions from non-communicable inflammatory skin diseases. *Nat Commun* 13, 7729 (2022). <https://doi.org/10.1038/s41467-022-35319-w>
- 3 Seremet, T. et al. Immune modules to guide diagnosis and personalized treatment of inflammatory skin diseases. *Nat Commun* 15, 10688 (2024). <https://doi.org/10.1038/s41467-024-54559-6>